# Low agreement and frequent invalid controls in two SARS-CoV-2 T-cell assays in people with compromised immune function

**Annette Audigé[1†], Alain Amstutz[2,3,4], Macé M. Schuurmans[5], Patrizia Amico[6], Dominique L. Braun[1,7], Marcel P. Stoeckle[8], Barbara Hasse[7], René Hage[5], Dominik Damm[5], Michael Tamm[9], Nicolas J. Mueller[7], Huldrych F. Günthard[1,7], Michael T. Koller[6,10], Christof M. Schönenberger[2], Alexandra Griessbach[2], Niklaus D. Labhardt[2], Roger D. Kouyos[1,7], Alexandra Trkola[1], Michael Huber[1], Katharina Kusejko[1,7], Heiner C. Bucher[2], Irene A. Abela[1,7], Matthias Briel[2,11], Frédérique Chammartin[2], Benjamin Speich[2]\*, the Swiss HIV Cohort Study, and the Swiss Transplant Cohort Study[¶]**

1 Institute of Medical Virology, University of Zurich, Zurich, Switzerland, 2 Division of Clinical Epidemiology, Department of Clinical Research, University Hospital Basel, University of Basel, Basel, Switzerland, 3 Oslo Center for Biostatistics and Epidemiology (OCBE), Oslo University Hospital, University of Oslo, Oslo, Norway, 4 Bristol Medical School, University of Bristol, Bristol, United Kingdom, 5 Division of Pulmonology, University Hospital Zurich, Zurich, Switzerland, 6 Clinic for Transplantation Immunology and Nephrology, University Hospital Basel, Basel, Switzerland, 7 Department of Infectious Diseases and Hospital Epidemiology, University Hospital Zurich, Zurich, Switzerland, 8 Division of Infectious Diseases and Hospital Epidemiology, University Hospital Basel, University of Basel, Basel, Switzerland, 9 Clinic of Respiratory Medicine and Pulmonary Cell Research, University Hospital Basel, Basel, Switzerland, 10 Swiss Transplant Cohort Study, University Hospital Basel, Basel, Switzerland, 11 Department of Health Research Methods, Evidence, and Impact, McMaster University, Hamilton, Canada

† Deceased.
¶ Membership of The Swiss HIV Cohort Study, and the Swiss Transplant Cohort Study is provided in the Acknowledgments.
\* benjamin.speich@usb.ch

## Abstract

T-cell response plays an important role in SARS-CoV-2 immunogenicity. For people living with HIV (PWH) and solid organ transplant (SOT) recipients there is limited evidence on the reliability of commercially available T-cell tests. We assessed 173 blood samples from 81 participants (62 samples from 35 PWH; 111 samples from 46 SOT recipients [lung and kidney]) with two commercial SARS-CoV-2 Interferon-γ (IFN-γ) release assays (IGRA; SARS-CoV-2 IGRA by Euroimmun, and IGRA SARS-CoV-2 by Roche). The reliability between the tests was judged as low (Cohen's kappa [κ] = 0.20; overall percent agreement [OPA] = 66%). A high proportion of tests were invalid (22% Euroimmun; 8% Roche). When excluding these invalid tests, the agreement was higher (κ = 0.43; OPA = 90%). The low reliability between the two T-cell tests indicates that results should be interpreted with caution in SOT recipients and PWH and that SARS-CoV-2 T-cell tests need to be optimized and further validated for use in vulnerable patient populations.

**Data Availability Statement:** As requested by the journal, we provide the data set for the diagnostic comparison (see supplementary excel file).

**Funding:** The vaccine study in which the samples for the T-cell assays were collected (i.e. COVERALL-3) was funded by Moderna. Roche provided all reagents for the T-cell assay Elecsys® IGRA SARS-CoV-2. The set-up of the study platform (i.e. COVERALL) was funded by the Swiss National Science Foundation (grant # 31CA30_196245). The Swiss HIV Cohort Study (SHCS) and the Swiss Transplant Cohort Study (STCS) are funded by the Swiss National Science Foundation (SHCS: grant #177499 and #201369, STCS: grant #33CS30_177522). Alain Amstutz received his salary for the duration of this project from the Junior Research Fund of the University of Basel. Christof Manuel Schönenberger received his salary for the duration of this project from the Swiss National Science Foundation (grant # 323530_221860) and the Janggen Pöhn Foundation. The funders had no role in data collection, analysis, and preparation of the manuscript. Before submission, Moderna and Roche had the right to read the manuscript and make suggestions, but the study team was not obliged to accept suggestions and the Funders were not involved in the final decision to submit to the journal.

**Competing interests:** Benjamin Speich and Matthias Briel received unrestricted grants from Moderna (2021/22) for the conduct of the COVERALL-2 and COVERALL-3 study. Heiner C. Bucher received in the 36 months prior to the submission of this manuscript one grant from Gilead that was not related to this project. Heiner C. Bucher served as the president of the 'Association contre le HIV et autres infections transmissibles' until June 2022. In this role he received support for the Swiss HIV Cohort Study from ViiV Healthcare, Gilead, BMS, and MSD. Alexandra Trkola received unrestricted research funding from the Swiss National Science Foundation, the Swiss HIV Cohort Study, Gilead Sciences and Novartis not related to this study. Dominique L. Braun received honoraria for advisory boards from the companies Gilead, MSD, Pfizer, AstraZeneca and ViiV outside of the study. Irene A. Abela received a research grant from Gilead sciences and honoraria for advisory boards from the companies Moderna and AstraZeneca. Huldrych F. Günthard, outside of this study, reports grants from the Swiss National Science Foundation, National Institutes of Health (NIH), and the Swiss HIV Cohort Study, unrestricted research grants from the Bill and Melinda Gates Foundation, Gilead Sciences, ViiV

## Introduction

By the end of 2020, initial findings from SARS-CoV-2 vaccine trials became available, indicating that the vaccines were over 90% effective in temporarily preventing COVID-19 [1, 2]. Given uncertainties about the vaccines' efficacy in individuals with compromised immune systems, our research group initiated the Corona VaccinE tRiAL pLatform (COVERALL) within the Swiss HIV Cohort Study (SHCS) and the Swiss Transplant Cohort Study (STCS) [3–5]. Since the vaccines were designed to stimulate an antibody response against the SARS-CoV-2 spike (S1) protein receptor binding domain, the first two COVERALL studies assessed whether people with HIV (PWH) and solid organ transplant (SOT) recipients developed adequate antibody response post vaccination [6, 7]. The emergence of highly mutable variants like Delta and Omicron compromised the neutralizing capability of antibodies [8], emphasizing the growing importance of the T-cell response [8–11].

Therefore, not only the antibody response but also T-cell response in PWH and SOT recipients (i.e. lung and kidney transplants), following a bivalent mRNA SARS-CoV-2 booster vaccination, was evaluated [12]. During this evaluation we observed differences amongst test results when using different commercially available T-cell tests. Since the reliability of the T-cell tests in people with compromised immune function is unclear, we therefore aimed to assess the agreement of two commercially available T-cell tests.

## Methods

### Study design

The methods and main results of the COVERALL-3 study have been published separately [12]. In brief, cohort participants from the SHCS and the STCS who had previously already revived the "basic immunization" SARS-CoV-2 vaccination (e.g. two doses of Spikevax from Moderna or two doses of Comirnaty from Pfizer-BioNtech) and who received the bivalent vaccine (mRNA-1273.214 or BA.1–adapted BNT162b2) were recruited from the 27th of October 2022 until the 24th of January 2023. They provided whole blood samples before vaccination (i.e. baseline), and at 4-weeks, 8-weeks, and 6-months post vaccination to measure the antibody response (full eligibility criteria in Supplement 1). A subset of participants provided 8 ml heparinized blood at baseline, 4-weeks, and 6-months to assess the T-cell response. Samples were promptly transported within 6 hours to the Institute of Medical Virology in Zürich where two commercial SARS-CoV-2 Interferon-γ (IFN-γ) release assays (IGRA) by Euroimmun, and Roche were conducted (i.e. SARS-CoV-2 IGRA by Euroimmun, and IGRA SARS-CoV-2 by Roche). The COVERALL-3 study received approval from the ethics committee Nordwest- and Zentralschweiz, Switzerland (BASEC Nr. 2022–01760), and the study protocol is accessible in a trial registry (https://clinicaltrials.gov/ct2/show/NCT04805125). Participants provided written consent for study participation and further utilization of collected biological samples.

### Laboratory measurements

Heparinized blood samples were processed within 16 hours after blood withdrawal. The two following SARS-CoV-2 IGRA were used for analyzing the T-cell response following the manufacturers' instructions: i) the quantitative SARS-CoV-2 IGRA by Euroimmun which combines the Quant-T-Cell SARS-CoV-2 kit for the T-cell stimulation and the Quant-T-Cell ELISA for measuring the released IFN-γ (Euroimmun Medizinische Labordiagnostica, Lübeck, Germany); and ii) the qualitative IGRA SARS-CoV-2 by Roche which combines the cobas IGRA SARS-CoV-2 tubes for the T-cell stimulation and the Elecsys IGRA SARS-CoV-2, an

Healthcare and Yvonne Jacob Foundation, personal fees from consulting or advisory boards or data safety monitoring boards for Merck, Gilead Sciences, ViiV Healthcare, Janssen, Johnson and Johnson, GSK and Novartis. Huldrych F. Günthard's institution received money for participation in the following clinical COVID-19 studies: 540-7773/5774 (Gilead), TICO (ACTIV-3, INSIGHT/NIH), and the Morningsky study (Roche). DLB reports honoraria for advisory boards, lectures, and travel grants from the companies Gilead, MSD and ViiV outside of the submitted work. Nicolas J. Mueller reports honoraria for advisory boards and travel grants from the companies Gilead, Biotest, Takeda outside of the submitted work. Roger D. Kouyos reports grants from the Swiss National Science Foundation, National Institutes of Health (NIH), the Swiss HIV Cohort Study, and Gilead Sciences (all outside of this study). All other authors have declared that no competing interests exist.

electrochemiluminescence immunoassay (ECLIA), for measuring the released IFN-γ (Roche Diagnostics, Rotkreuz, Switzerland).

The kits for T-cell stimulation from both manufacturers consist of three stimulation tubes per whole-blood sample: 1) no T-cell stimulation, for determination of the individual IFN-γ background; 2) specific T-cell stimulation (Euroimmun: antigens based on the S1-domain of the SARS-CoV-2 spike protein; Roche: peptides derived from structural and non-structural proteins of SARS-CoV-2); and 3) unspecific T-cell stimulation by means of a mitogen, for control of the stimulation ability. The Euroimmun assay uses antigens based on the S1-domain of the SARS-CoV-2 spike protein; the Roche assay uses more than 180 antigens derived from structural (spike, membrane and nucleocapsid) and non-structural proteins of SARS-CoV-2.

Data generated with the immunoassays were analysed applying the validation criteria for the negative and positive controls as defined by the manufacturers (Euroimmun: positive control ≥400 mIU/ml, negative control ≤400 mIU/ml; Roche: positive control ≥1 IU/ml, negative control ≤0.3 IU/ml). Samples with a non-valid or indeterminate negative or positive stimulation control were labelled "invalid". Results of the specific stimulation were classified according to the manufacturers' recommendations either as positive (>200 mIU/ml), negative (<100 mIU/ml), borderline (≥100 mIU/ml to ≤200 mIU/ml), or invalid for the Euroimmun assay, and as positive (≥0.013IU/ml), negative (<0.013 IU/ml), or invalid for the Roche assay (for comparability, non-reactive and reactive results with the Roche test were labelled negative and positive, respectively).

## Data analysis

We used Cohen's kappa to assess the agreement between the two T-cell tests. Cohen's kappa takes also into account the agreement by chance which can be high when the proportion of true negative or true positive is high (see references for further details [13, 14]). In addition, we also present the simple overall percentage agreement (OPA) which does not adjust for chance agreement. The overall percentage agreement simply assesses the number of agreeing test results for both diagnostic tests divided by all test results. For the interpretation of the kappa we used the following scale as proposed by McHugh [13]: κ = 0–0.20 indicates no agreement; κ = 0.21–0.39 suggests minimal agreement; κ = 0.40–0.59 indicates weak agreement; κ = 0.60–0.70 signifies moderate agreement; κ = 0.80–0.90 implies strong agreement; and κ>0.90 denotes almost perfect agreement. In the main analysis, we created a 3x3 table (comprising positive T-cell response, negative T-cell response, and invalid tests) in which borderline results from the Euroimmun test were categorized as positive results. In contrast to other studies [15, 16] we included invalid in our analyses as also these results need to be communicated to clinicians and patients in routine practice. However, we conducted two sensitivity analyses, in which we (i) excluded samples classified as invalid by either test (while treating borderline results from the Euroimmun test as positive), and (ii) treated borderline results from the Euroimmun test as negative. Furthermore, we explored patient characteristics associated with invalid test results, conducted analyses including only PWH or SOT recipients, and assessed each time point separately.

## Results

Between October 2022 and September 2023, 81 of the 174 COVERALL-3 [12] study participants provided blood samples to assess the T-cell response. Among these 81 participants, 173 samples were available for testing with both T-cell tests (baseline [n = 40], 4-weeks follow-up [n = 74] and 6-months follow-up [n = 59]). Of these 173 samples, 62 were from PWH (20 with CD4 cell counts <350 cells/μl; 42 with CD4 cell counts ≥350 cells/μl) and 111 from SOT

recipients (80 lung transplants; 31 kidney transplants). Baseline characteristics of the 81 participants are presented in S1 Table in S1 File.

Using the Euroimmun T-cell test, 105 of the 173 (60.7%) samples exhibited a positive T-cell response, 12 (6.9%) were borderline, 18 (10.4%) showed no response, and 38 (22.0%) were invalid (Table 1). Using the Roche T-cell test, 139 (80.3%) samples had a positive response, 20 (11.6%) showed no response, and 14 (8.1%) were invalid (Table 1). Among PWH, 2 (3.2%) had an invalid result using the Euroimmun test and 8 (12.9%) with the Roche test. Among SOT recipients, 36 (32.4%) had an invalid result with the Euroimmun test and 6 (5.4%) with the Roche test (Table 1).

Classifying borderline results from the Euroimmun test as positive, kappa was minimal (κ = 0.20) and the OPA was 66% (Table 1; S2 Table in S1 File). When stratifying by PWH (κ = 0.14; OPA: 85%) and SOT recipients (κ = 0.18; OPA: 54%), higher OPA for PWH and low kappas were observed (Table 1). The overall kappa decreased further by treating borderline results from the Euroimmun test as negative (κ = 0.14; OPA = 59%; S3 Table in S1 File). The agreement was similarly low when separately assessing samples from baseline, 4-weeks, and

**Table 1. Agreement between the SARS-CoV-2 IGRA by Euroimmun and the IGRA SARS-CoV-2 by Roche using the cut-off categories as provided by the manufacturers.**

**People living with HIV and solid organ transplant recipients combined**

| SARS-CoV-2 IGRA by Euroimmun | | IGRA SARS-CoV-2 by Roche | | | |
|---|---|---|---|---|---|
| | | Positive | Negative | Invalid | Total |
| | Positive | 93 | 1 | 11 | 105 (60.7%) |
| | Borderline* | 12 | 0 | 0 | 12 (6.9%) |
| | Negative | 12 | 6 | 0 | 18 (10.4%) |
| | Invalid | 22 | 13 | 3 | 38 (22.0%) |
| | Total | 139 (80.3%) | 20 (11.6%) | 14 (8.1%) | 173 (100.0%) |

κ = 0.20; overall percent agreement: 66%; expected agreement 57%.*

**People living with HIV from the Swiss HIV Cohort Study**

| SARS-CoV-2 IGRA by Euroimmun | | IGRA SARS-CoV-2 by Roche | | | |
|---|---|---|---|---|---|
| | | Positive | Negative | Invalid | Total |
| | Positive | 49 | 1 | 7 | 57 (91.9%) |
| | Borderline* | 3 | 0 | 0 | 3 (4.8%) |
| | Negative | 0 | 0 | 0 | 0 (0.0%) |
| | Invalid | 1 | 0 | 1 | 2 (3.2%) |
| | Total | 53 (85.5%) | 1 (1.6%) | 8 (12.9%) | 62 (100.0%) |

κ = 0.14; overall percent agreement: 85%; expected agreement: 83%.*

**Solid organ transplant recipients from the Swiss Transplant Cohort Study**

| SARS-CoV-2 IGRA by Euroimmun | | IGRA SARS-CoV-2 by Roche | | | |
|---|---|---|---|---|---|
| | | Positive | Negative | Invalid | Total |
| | Positive | 44 | 0 | 4 | 48 (43.2%) |
| | Borderline* | 9 | 0 | 0 | 9 (8.1%) |
| | Negative | 12 | 6 | 0 | 18 (16.2%) |
| | Invalid | 21 | 13 | 2 | 36 (32.4%) |
| | Total | 86 (77.5%) | 19 (17.1%) | 6 (5.4%) | 111 (100.0%) |

κ = 0.18; overall percent agreement: 54%; expected agreement: 44%.*

*For calculating κ and agreements, borderline results from the Euroimmun test were classified as a positive T-cell response.

Abbreviations: IGRA = Interferon-γ release assay

6-months (S4 Table in S1 File). Excluding all invalid results from either test, a high OPA (90%), but a weak agreement based on kappa was observed (Table 2).

For both tests, IFN-γ concentrations of the positive controls were lower for SOT recipients compared to PWH (Fig 1). For two SOT recipients, all samples from all three timepoints were invalid using the Euroimmun test (S1 Fig in S1 File). Detailed analyses showed a median of 426 days since organ transplantation (Interquartile Range [IQR] 351–500) for these SOT recipients, in contrast to 2,822 days (IQR: 1,072–3,315) for SOT recipients with no invalid Euroimmun test (S5 Table in S1 File). Positive controls that were invalid with the Euroimmun test had low positive control values with the Roche test (S2 Fig in S1 File). A description of all invalid test results is provided in S6 Table in S1 File. Especially, most invalid results of the Euroimmun test were invalid positive controls in SOT recipients.

## Discussion

Ideally two commercially available tests measuring the SARS-CoV-2 T-cell response in the same blood sample should come to the same result in order to be trustworthy and so that clinicians can make an informed decision together with patients. Our study showed low agreement based on kappa when assessing the T-cell response in individuals with compromised immune function with the two commercially available SARS-CoV-2 IGRA T-cell tests by Euroimmun and Roche. As case distribution between positive and negative was uneven (especially amongst PWH), a low kappa did not necessarily mean low OPA (kappa paradox [17]). However, the high OPA was also strongly driven by the high expected agreement (i.e. agreement by chance), and the low kappa confirmed that the tests had difficulties in distinguishing the few negative and the invalid results. In agreement with our results, Carretero and colleagues found a low kappa (κ = 0.29) for the T-cell tests by Euroimmun and Roche amongst 50 kidney patients

**Table 2. Sensitivity analysis, assessing the agreement between the SARS-CoV-2 IGRA by Euroimmun and the IGRA SARS-CoV-2 by Roche when dropping samples that were invalid with either test.**

**People living with HIV and solid organ transplant recipients combined**

| SARS-CoV-2 IGRA by Euroimmun | | IGRA SARS-CoV-2 by Roche | | |
|---|---|---|---|---|
| | | Positive | Negative | Total |
| | Positive | 105 | 1 | 106 (85.5%) |
| | Negative | 12 | 6 | 18 (14.5%) |
| | Total | 117 (94.4%) | 7 (5.6%) | 124 (100.0%) |

κ = 0.43; overall percent agreement: 90%; expected agreement 81%

**People living with HIV from the Swiss HIV Cohort Study**

| SARS-CoV-2 IGRA by Euroimmun | | | | |
|---|---|---|---|---|
| | | Positive | Negative | Total |
| | Positive | 52 | 1 | 53 (100.0%) |
| | Negative | 0 | 0 | 0 (0.0%) |
| | Total | 52 (98.1%) | 1 (1.9%) | 53 (100.0%) |

κ = -; overall percent agreement: 98%; expected agreement 98%

**Solid organ transplant recipients from the Swiss Transplant Cohort Study**

| SARS-CoV-2 IGRA by Euroimmun | | | | |
|---|---|---|---|---|
| | | Positive | Negative | Total |
| | Positive | 53 | 0 | 53 (74.6%) |
| | Negative | 12 | 6 | 18 (25.4%) |
| | Total | 65 (91.5%) | 6 (8.5%) | 71 (100.0%) |

κ = 0.43; overall percent agreement: 83%; expected agreement 70%

Borderline results from the Euroimmun test were classified as a positive T-cell response and invalid results from either test were dropped.

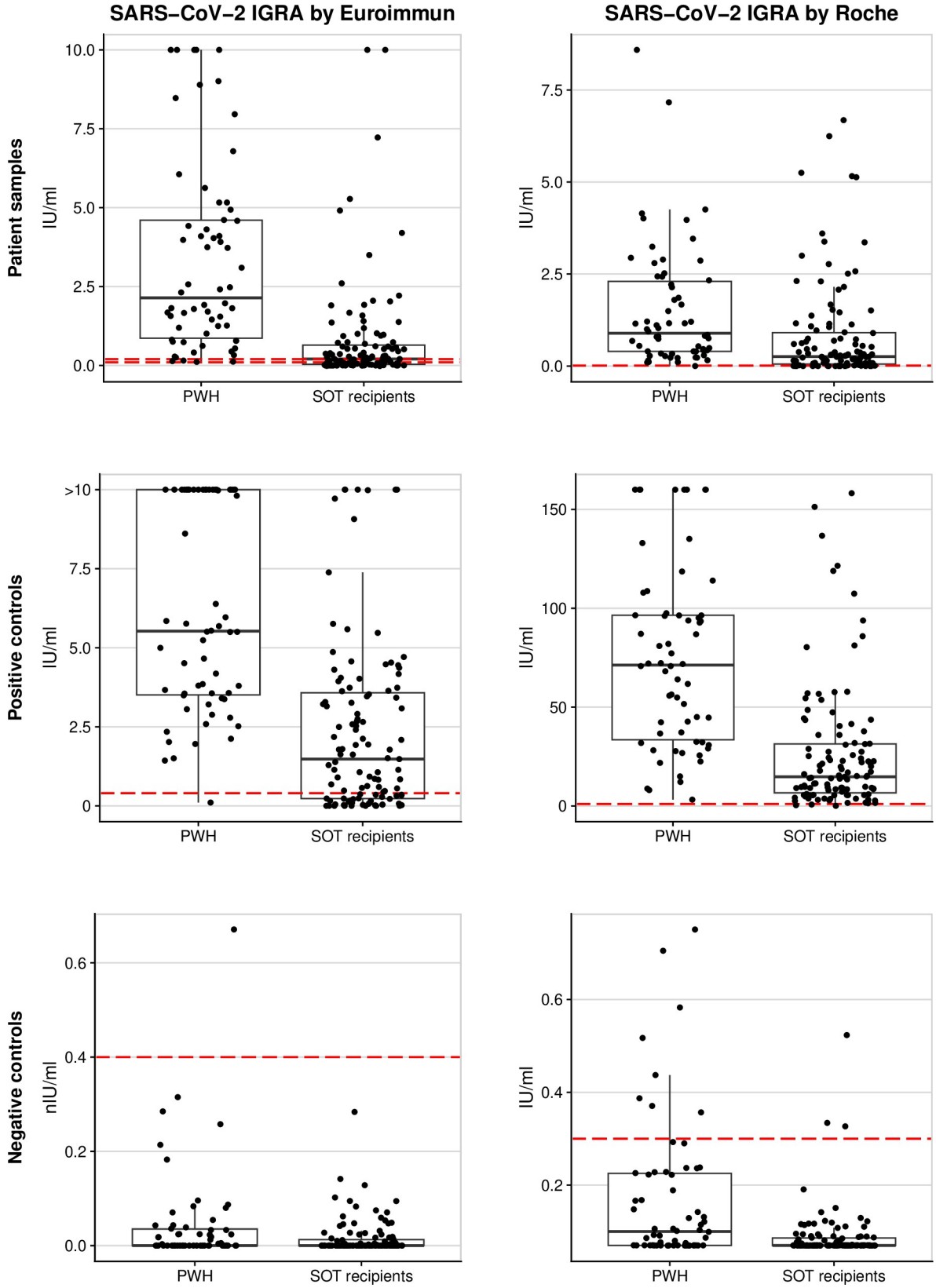

**Fig 1. Interferon-γ concentrations of samples and the positive and negative controls using either the Euroimmun or the Roche T-cell test.** Red lines indicate the thresholds for positivity (and borderline) for patients samples and for validity of the positive and negative controls, respectively. Abbreviations: IGRA = Interferon-γ release assays; PWH = People with HIV; SOT = Solid organ transplant.

(n = 24 kidney transplant recipients; n = 26 chronic kidney disease) [15]. In contrast to these results, a study involving 89 healthcare workers found substantial agreement among four T-cell assays (two ELISA-based IGRA and two IFN-γ ELISPOT assays) [18]. The authors observed a stronger correlation between assays within the same platform (IGRA or ELISPOT) than between different platforms. The measurement of released IFN-γ either in supernatant (ELISA-based IGRA), or immobilized (IFN-γ ELISPOT) could contribute to this difference. Of note, the Euroimmun IGRA and the QuantiFERON IGRA (Qiagen) are both ELISAs, while the IGRA by Roche used in our study is an electrochemiluminescence immunoassay. Additionally, the discrepancy between the Euroimmun and Roche IGRAs may also stem from differences in the type of antigen utilized. Natural SARS-CoV-2 infection induces cellular immunity not only towards spike (the Euroimmun antigen is solely based on the S1-domain of the spike protein), but also towards other viral proteins (the Roche antigens are peptides derived from several structural and non-structural proteins).

The potential for invalid results due to impaired T-cell activity or low T-cell numbers, among patients with different levels of immunosuppression, represents an additional challenge for T-cell response assessment. A large proportion (22%) of Euroimmun SARS-CoV-2 IGRA tests, mainly from participants having recently received a SOT, were invalid due to an invalid positive control. This might be a result of the more immunosuppressed state of this patient group compared to PWH, in combination with the antigen used in the Euroimmun test. This finding is in line with a study conducted by Saad-Albichr et al., where 20% of kidney transplant recipients had invalid Euroimmun T-cell results [16]. Conversely, when using the IGRA SARS-CoV-2 by Roche to assess the T-cell response, the proportion of invalid results in our study was 8% (14/173), mainly due to an invalid negative control (79%; 11/14). The difference in the proportion of invalid positive controls between the two tests may stem from differences in the type and concentration of the mitogens/stimulants in the positive controls, details of which are undisclosed by the manufacturers. Meanwhile, discrepancies in the proportion of invalid negative controls are likely due to varying cut-offs established by the manufacturers. These data underscore the necessity for optimizing T-cell assays for patients with different levels of immunosuppression.

The following limitations are worth mentioning: Firstly, we found low agreement between the two T-cell tests. However, due to the absence of a gold standard we cannot make any statement about the true performance of the individual tests. While some assessments showed high sensitivity and specificity of the SARS-CoV-2 IGRA by Euroimmun in the general population [19, 20], we are aware of only one published assessment of the IGRA SARS-CoV-2 by Roche which was conducted in immunosuppressed patients and excluded invalid results from the analyses [15]). Secondly, with 173 samples from 81 participants our sample size is limited. Hence, especially the sub-group analyses should be interpreted with caution.

In conclusion, our results indicate low agreement between two commercial SARS-CoV-2 T-cell tests in PWH and SOT recipients, raising concerns regarding the reliability of these tests in individuals with compromised immune function. Especially the Euroimmun test showed a high proportion of invalid test results in SOT recipients due to an insufficient positive control. Hence, optimizing and validating SARS-CoV-2 T-cell tests in vulnerable populations, for whom such tests are of particular importance, is crucial.

## Supporting information

**S1 Dataset.**
(XLSX)

**S1 File.**
(DOCX)

## Acknowledgments

We are grateful to all involved study staff at the local centers in Basel (Maria Pascarella, Louise Seiler), Zürich (Daniela Gsell, Katia Dettling, Laura Tschuor, Christina Grube, Flurina Brunschweiler, Christine Schneider, Andrea Wallensteiner, Daniel Götsch, Andrea Macedo, Esther Göldi, Alina Imoli), and the Institute of Medical Virology, University of Zürich. Most importantly, we thank all study participants.

**Members of the Swiss HIV Cohort Study**

Abela I, Aebi-Popp K, Anagnostopoulos A, Battegay M, Bernasconi E, Braun DL, Bucher HC, Calmy A, Cavassini M, Ciuffi A, Dollenmaier G, Egger M, Elzi L, Fehr J, Fellay J, Furrer H, Fux CA, Günthard HF (President of the SHCS), Hachfeld A, Haerry D (deputy of "Positive Council"), Hasse B, Hirsch HH, Hoffmann M, Hösli I, Huber M, Jackson-Perry D (patient representatives), Kahlert CR (Chairman of the Mother & Child Substudy), Kaiser L, Keiser O, Klimkait T, Kouyos RD, Kovari H, Kusejko K (Head of Data Centre), Labhardt N, Leuzinger K, Martinez de Tejada B, Marzolini C, Metzner KJ, Müller N, Nemeth J, Nicca D, Notter J, Paioni P, Pantaleo G, Perreau M, Rauch A (Chairman of the Scientific Board), Salazar-Vizcaya L, Schmid P, Speck R, Stöckle M (Chairman of the Clinical and Laboratory Committee), Tarr P, Trkola A, Wandeler G, Weisser M, Yerly S.

**Members of the Swiss Transplant Cohort Study**

The members of the Swiss Transplant Cohort Study: Patrizia Amico, John-David Aubert, Vanessa Banz, Sonja Beckmann, Guido Beldi, Christoph Berger, Ekaterine Berishvili, Annalisa Berzigotti, Isabelle Binet, Pierre-Yves Bochud, Sanda Branca, Heiner C. Bucher, Emmanuelle Catana, Anne Cairoli, Yves Chalandon, Sabina De Geest, Olivier De Rougemont, Sophie De Seigneux, Michael Dickenmann, Joëlle Lynn Dreifuss, Michel Duchosal, Thomas Fehr, Sylvie Ferrari-Lacraz, Christian Garzoni, Déla Golshayan, Nicolas Goossens, Fadi Haidar, Jörg Halter, Dominik Heim, Christoph Hess, Sven Hillinger, Hans H Hirsch, Patricia Hirt, Linard Hoessly, Günther Hofbauer, Uyen Huynh-Do, Franz Immer, Michael Koller, Bettina Laesser, Frédéric Lamoth, Roger Lehmann, Alexander Leichtle, Oriol Manuel, Hans-Peter Marti, Michele Martinelli, Valérie McLin, Katell Mellac, Aurélia Merçay, Karin Mettler, Nicolas J Mueller, Ulrike Müller-Arndt, Beat Müllhaupt, Mirjam Nägeli, Graziano Oldani, Manuel Pascual, Jakob Passweg, Rosemarie Pazeller, Klara Posfay-Barbe, Juliane Rick, Anne Rosselet, Simona Rossi, Silvia Rothlin, Frank Ruschitzka, Thomas Schachtner, Stefan Schaub, Alexandra Scherrer, Aurelia Schnyder, Macé Schuurmans, Simon Schwab, Thierry Sengstag, Federico Simonetta, Susanne Stampf, Jürg Steiger, Guido Stirnimann, Ueli Stürzinger, Christian Van Delden, Jean-Pierre Venetz, Jean Villard, Julien Vionnet, Madeleine Wick, Markus Wilhelm, Patrick Yerly.

## Author Contributions

**Conceptualization:** Annette Audigé, Alain Amstutz, Christof M. Schönenberger, Alexandra Griessbach, Niklaus D. Labhardt, Heiner C. Bucher, Irene A. Abela, Matthias Briel, Frédérique Chammartin, Benjamin Speich.

**Data curation:** Annette Audigé, Macé M. Schuurmans, Patrizia Amico, Dominique L. Braun, Marcel P. Stoeckle, Barbara Hasse, René Hage, Dominik Damm, Michael Tamm, Nicolas J. Mueller, Huldrych F. Günthard, Michael T. Koller.

**Formal analysis:** Michael Huber, Frédérique Chammartin, Benjamin Speich.

**Methodology:** Alain Amstutz, Benjamin Speich.

**Project administration:** Alexandra Griessbach, Roger D. Kouyos, Katharina Kusejko, Benjamin Speich.

**Resources:** Annette Audigé, Katharina Kusejko.

**Software:** Katharina Kusejko.

**Supervision:** Matthias Briel.

**Validation:** Irene A. Abela, Frédérique Chammartin.

**Visualization:** Christof M. Schönenberger, Alexandra Trkola, Michael Huber, Irene A. Abela, Frédérique Chammartin, Benjamin Speich.

**Writing – original draft:** Annette Audigé, Benjamin Speich.

**Writing – review & editing:** Alain Amstutz, Macé M. Schuurmans, Patrizia Amico, Dominique L. Braun, Marcel P. Stoeckle, Barbara Hasse, René Hage, Dominik Damm, Michael Tamm, Nicolas J. Mueller, Huldrych F. Günthard, Michael T. Koller, Christof M. Schönenberger, Alexandra Griessbach, Niklaus D. Labhardt, Roger D. Kouyos, Alexandra Trkola, Michael Huber, Katharina Kusejko, Heiner C. Bucher, Irene A. Abela, Matthias Briel, Frédérique Chammartin.

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
