## [Decision Letter · Decision Letter 0]

12 Nov 2024

PONE-D-24-37872Low agreement and frequent invalid controls in two SARS-CoV-2 T-cell assays in people with compromised immune functionPLOS ONE

Dear Dr. Speich,

Thank you for submitting your manuscript to PLOS ONE. After careful consideration, we feel that it has merit but does not fully meet PLOS ONE’s publication criteria as it currently stands. Therefore, we invite you to submit a revised version of the manuscript that addresses the points raised during the review process.

We look forward to receiving your revised manuscript.

Kind regards,

Mao-Shui Wang

Academic Editor

PLOS ONE

Journal Requirements:

2. In the online submission form, you indicated that data are available from corresponding author on reasonable request.

Additional Editor Comments:

Please respond to the reviewers' comments and address the concerns they raised.

Reviewers' comments:

Reviewer's Responses to Questions

**Comments to the Author**

1. Is the manuscript technically sound, and do the data support the conclusions?

Reviewer #1: Yes

Reviewer #2: Yes

Reviewer #3: Partly

2. Has the statistical analysis been performed appropriately and rigorously? 

Reviewer #1: Yes

Reviewer #2: Yes

Reviewer #3: N/A

3. Have the authors made all data underlying the findings in their manuscript fully available?

Reviewer #1: No

Reviewer #2: Yes

Reviewer #3: Yes

4. Is the manuscript presented in an intelligible fashion and written in standard English?

Reviewer #1: Yes

Reviewer #2: Yes

Reviewer #3: Yes

5. Review Comments to the Author

Reviewer #1: Dear authors,

the authors have performed direct comparison of two commercially available tests for the assessment of SARS-CoV-2-specific cellular immunity in two different cohorts of immune impaired persons (people living with HIV, PWH and solid organ transplant recipients, SOT). They found low agreement of test results and high rates of invalid results with particular differences between the two patient groups and conclude that the tests need to be optimized especially for immune impaired patients which are most vulnerable. The manuscript is well written, and most of the data is presented clearly. However, I have some points which should be addressed to further improve clarity especially for readers, which are not familiar with the test principles in detail.

- Introduction: p.6, last sentence „However since the reliability of the T-cell tests in people with compromised immune function is unclear…“

Since this is the main fact, the study is based on, please explain in more detail (e.g. with IGRA performances concerning other pathogens like Mtb in these patient groups) or at least include adequate citations.

- Methods: p.7

Please state the validation criteria for negative and positive controls as well as the detection limits as defined by the manufacturers. This information is important to follow the conclusions adequately.

- Methods: p.8 (Data analysis)

In this section, the authors state, that borderline results from the Euroimmun test were categorized as positive for the 3x3 table comparison and as negative for sensitivity analysis. In contrast, in the footnote to table 2 (sensitivity analysis), it is stated that borderline results were categorized as positive. The latter would be more intuitive in my opinion and matches with the numbers given in the table. So, I guess, the „negative“ is a typo and should be "positive". If this is the case, please correct accordingly, if not, please explain why borderline results were treated as positive for 3x3 tables and as negative for sensitivity analyses.

- p.9, line 6: typo: PHW should be PWH

- Discussion: p.10

The authors compare IGRA with ELISPOT. I’d like to point out, that the ELISPOT assay is also an IGRA (interferon gamma release assay), as the IFNg is released by the cells either. The tests just differ in the way of readout (whole IFNg detected by ELISA in the supernatant vs IFNg immobilized via catch antibodies and detetcted as „spot“ on a plate). Please reword accordingly (e.g. comparing „ELISA-based IGRA“ with ELISPOT).

- Discussion: p.10

The discussion on the underlying causes for the differences in test performances should be expanded, at least for the following two points:

(i) ELISA and electrochemiluminescence immunoassay (both detect IFNg in the supernatant after stimulation) do not differ in the way/extent, ELISA and ELISPOT do (see above). Please comment.

(ii) the authors state that the discrepancy between the two tests „may also stem from differences in the type of antigen utilized“ without further explanation. I guess, this difference (EI: antigens against S1-domain of spike only vs. 189 peptides against various SARS-CoV-2 proteins) may be important, a. o. because SARS-CoV-2 infection induces cellular immunity not only towards spike (as it is the case for the applied vaccines) but also towards other viral proteins. Please expand the discussion concerning this point.

- Discussion: p.10, „However, due to the absence of a gold standard we cannot make any statement about true performance of the individual tests.“

This is true and an important point to mention. Nevertheless, can the authors give some information about expected reactivities in the two populations after mRNA-vaccination (from other studies with PWH and SOT)?

- Figure 1:

Please include the results of the antigen tubes as well to give an estimate on the respective distribution of the IFNg-levels.

- Table S1: Since the focus of the study is to compare two different tests in two different cohorts of immune impaired individuals (and not to compare two different vaccine regimens), I would recommend to compare PWH and SOT recipients in the two columns (with one line depicting the percentage of persons vaccinated with the Moderna or the Pfizer-BioNTech vaccine respectively). This would be more intuitive. Please consider revising accordingly.

Additionally, information on time since last immunizing event (SARS-CoV-2 vaccination or infection) before baseline measurement and (history of SARS-CoV-2 infection before each measurement) would be informative to assess the probability of a truely negative test result. Is this information available and can be included in table S1?

What is the rationale for including information on influenza vaccination? Please explain or delete.

Reviewer #2: Audigé et al. propose a study of T cell response to SARS-CoV2 in vaccinated immunocompromised patients: people living with HIV and solid organ transplant recipients.

The study is well presented and clear. However, several minor points needs to be clarified before publication:

- The threshold for positive response to the two tests need to be indicated

- The indeterminate results (or ‘invalid’) needs to be explained: is this a negative response to the positive control or a positive response to the negative control?

- The high proportion of negative results with the Euroimmun tests may be linked to the lack of SARS-Cov2 antigen stimulation (spike alone)

In the abstract, ‘vaccinated patients’ should be quoted. Also, ‘immunocompromised’ should replace ‘vulnerable’ patients.

Reviewer #3: 1. In the main text it is mentioned that data was collected in 2022-2023, thus it should be clarified that vaccination by 2 doses was not a primary vaccination regimen, as the majoriyt of patients received full doses of vaccine during the pandemic beginning

2.In the Methods section it is mentioned, that patients provided samples at several time points, but it seems like not all the patients provided blood samples at all the time points. Could you clarify why? did the timepoint of blood test from vaccination had an impact on the tests result?

3.Authors should explain how kappa and OPA are calculated, otherwise it is not clear why there is a discrepancy between the kappa and OPA

4.the main point that the methodology of the compared tests is different, thus it is hard to compare the results

5. How do you explain more invalid results with Roche in PWH and more invalid results of Euroimmun in SOT?

6. PLOS authors have the option to publish the peer review history of their article (what does this mean?). If published, this will include your full peer review and any attached files.

Reviewer #1: No

Reviewer #2: **Yes: **Amélie Guihot

Reviewer #3: No

---

## [Author Response · Author response to Decision Letter 0]

16 Dec 2024

PONE-D-24-37872

We would like to thank the editors and all reviewers for their thorough assessment of our manuscript and their helpful comments. We have implemented and clarified the comments to further strengthen our manuscript. Please see our responses below (in blue colour). Of note, all line numbers are referring to the track-changed version of our manuscript.

Reviewer 1

Dear authors,

the authors have performed direct comparison of two commercially available tests for the assessment of SARS-CoV-2-specific cellular immunity in two different cohorts of immune impaired persons (people living with HIV, PWH and solid organ transplant recipients, SOT). They found low agreement of test results and high rates of invalid results with particular differences between the two patient groups and conclude that the tests need to be optimized especially for immune impaired patients which are most vulnerable. The manuscript is well written, and most of the data is presented clearly. However, I have some points which should be addressed to further improve clarity especially for readers, which are not familiar with the test principles in detail.

Response: We thank reviewer 1 for the overall positive feedback and the constructive comments which we address below.

R1.1:- Introduction: p.6, last sentence „However since the reliability of the T-cell tests in people with compromised immune function is unclear…“

Since this is the main fact, the study is based on, please explain in more detail (e.g. with IGRA performances concerning other pathogens like Mtb in these patient groups) or at least include adequate citations.

Response: During the study where we assessed the T-cell response in PWH and SOT recipients after receiving a bivalent SARS-CoV-2 vaccine, we realised that the two used T-cell tests (from Euroimmun and Roche) are not always in agreement. As we did not find relevant literature (same patient population and same tests) we decided to conduct this study in which we assessed the agreement of these two T-cell tests. We have expanded on the rationale in the background section accordingly (see revised manuscript, lines 131-134)

R1.2:- Methods: p.7

Please state the validation criteria for negative and positive controls as well as the detection limits as defined by the manufacturers. This information is important to follow the conclusions adequately.

Response: We agree with the reviewer that limits for negative and positive controls as well as the detection limits are missing in the manuscript. We added the threshold values to the methods section (see revised manuscript, lines 172-173 and 175-179). The thresholds are also shown as red lines in Figure 1.

R1.3:- Methods: p.8 (Data analysis)

In this section, the authors state, that borderline results from the Euroimmun test were categorized as positive for the 3x3 table comparison and as negative for sensitivity analysis. In contrast, in the footnote to table 2 (sensitivity analysis), it is stated that borderline results were categorized as positive. The latter would be more intuitive in my opinion and matches with the numbers given in the table. So, I guess, the „negative“ is a typo and should be "positive". If this is the case, please correct accordingly, if not, please explain why borderline results were treated as positive for 3x3 tables and as negative for sensitivity analyses.

Response: We apologise that this was not entirely clear. In fact, we have conducted two sensitivity analyses. Reviewer 1 is correct that for the sensitivity analyses in which invalid results were excluded, the borderline test results from Euroimmun were treated as positive. The results from the second sensitivity analysis in which we treated borderline results from the Euroimmun test as negative are presented in the Table S3. In the revised manuscript we have clarified in the methods section that we have conducted two sensitivity analyses: “However, we conducted two sensitivity analyses, in which we (i) excluded samples classified as invalid by either test (while treating borderline results from the Euroimmun test as positive), and (ii) treated borderline results from the Euroimmun test as negative.” (see revised manuscript, lines 196-200).

R1.4:- p.9, line 6: typo: PHW should be PWH

Response: Thank you for spotting this typo. We corrected it as suggested (see revised manuscript, line 216).

R1.5:- Discussion: p.10

The authors compare IGRA with ELISPOT. I’d like to point out, that the ELISPOT assay is also an IGRA (interferon gamma release assay), as the IFNg is released by the cells either. The tests just differ in the way of readout (whole IFNg detected by ELISA in the supernatant vs IFNg immobilized via catch antibodies and detected as „spot“ on a plate). Please reword accordingly (e.g. comparing „ELISA-based IGRA“ with ELISPOT).

Response: We agree with the reviewer that both assays measure the release of interferon gamma, one in supernatant, the other immobilized. We therefore reworded as suggested as “ELISA-based IGRA and IFN-γ ELISPOT” (line 246).

R1.6:- Discussion: p.10

The discussion on the underlying causes for the differences in test performances should be expanded, at least for the following two points:

(i) ELISA and electrochemiluminescence immunoassay (both detect IFNg in the supernatant after stimulation) do not differ in the way/extent, ELISA and ELISPOT do (see above). Please comment.

(ii) the authors state that the discrepancy between the two tests „may also stem from differences in the type of antigen utilized“ without further explanation. I guess, this difference (EI: antigens against S1-domain of spike only vs. 189 peptides against various SARS-CoV-2 proteins) may be important, a. o. because SARS-CoV-2 infection induces cellular immunity not only towards spike (as it is the case for the applied vaccines) but also towards other viral proteins. Please expand the discussion concerning this point.

Response: We thank the reviewer for these suggestions. We added a sentence on the two different detection methods “The measurement of released IFN-γ either in supernatant (ELISA-based IGRA), or immobilized (IFN-γ ELISPOT) could contribute to this difference” (lines 248-249).

We also expanded the discussion the different antigens used by stating that “Natural SARS-CoV-2 infection induces cellular immunity not only towards spike (the Euroimmun antigen is solely based on the S1-domain of the spike protein), but also towards other viral proteins (the Roche antigens are peptides derived from several structural and non-structural proteins)” (lines 252-255).

R1.7:- Discussion: p.10, „However, due to the absence of a gold standard we cannot make any statement about true performance of the individual tests.“

This is true and an important point to mention. Nevertheless, can the authors give some information about expected reactivities in the two populations after mRNA-vaccination (from other studies with PWH and SOT)?

Response: We have added now more context and refer to recent studies which have assessed the performance of the two diagnostic tests. “While some assessments showed high sensitivity and specificity of the SARS-CoV-2 IGRA by Euroimmun [19-20] in the general population, we are aware of only one published assessment of the IGRA SARS-CoV-2 by Roche which was conducted in immunosuppressed patients and excluded invalid results from the analyses [15].” (see revised manuscript, lines 275-278).

R1.8:- Figure 1:

Please include the results of the antigen tubes as well to give an estimate on the respective distribution of the IFNg-levels.

Response: As the author suggested, we included the results of the patient samples in Figure 1. Figure 1 now has three rows for patient samples, positive controls and negative controls.

R1.9:- Table S1: Since the focus of the study is to compare two different tests in two different cohorts of immune impaired individuals (and not to compare two different vaccine regimens), I would recommend to compare PWH and SOT recipients in the two columns (with one line depicting the percentage of persons vaccinated with the Moderna or the Pfizer-BioNTech vaccine respectively). This would be more intuitive. Please consider revising accordingly.

Additionally, information on time since last immunizing event (SARS-CoV-2 vaccination or infection) before baseline measurement and (history of SARS-CoV-2 infection before each measurement) would be informative to assess the probability of a truely negative test result. Is this information available and can be included in table S1?

What is the rationale for including information on influenza vaccination? Please explain or delete.

Response: We completely agree with the reviewer and have therefore generated a new Table S1 following the recommendations from Reviewer 1. In brief, we present characteristics now stratified by PWH and SOT patients instead of vaccination received, we have dropped the variable “flu vaccine” and “days since last SARS-CoV-2 vaccination before receiving the bivalent SARS-CoV-2 vaccine”. Unfortunately, we do not have information about previous SARS-CoV-2 infections (only antibody test to the nucleocapsid protein as a proxy). However, please note that the agreement of tests did not improve after receiving the bivalent SARS-CoV-2 vaccine (Table S4).

Reviewer 2

Audigé et al. propose a study of T cell response to SARS-CoV2 in vaccinated immunocompromised patients: people living with HIV and solid organ transplant recipients.

The study is well presented and clear. However, several minor points needs to be clarified before publication:

Response: We thank reviewer 2 for the careful assessment and the positive feedback.

R2.1:- The threshold for positive response to the two tests need to be indicated

Response: We thank the reviewer for this comment. As already mentioned in the response to reviewer 1 (see R.1.2 above), we added the threshold values to the methods section (see revised manuscript, lines 172-173 and 175-179). The thresholds are also shown as red lines in Figure 1.

R2.2:- The indeterminate results (or ‘invalid’) needs to be explained: is this a negative response to the positive control or a positive response to the negative control?

Response: We explain the invalid results in terms of invalid negative or positive controls, respectively, in Table S6. In the revised version, we expanded this table (see also answer to reviewer #3 point R3.5 below).

R2.3:- The high proportion of negative results with the Euroimmun tests may be linked to the lack of SARS-Cov2 antigen stimulation (spike alone)

Response: We thank the reviewer for this comment. As already mentioned in the response to reviewer 1 (see point 1.6 above), we expanded the discussion on the different antigens used in both tests (lines 252-255).

R2.4: In the abstract, ‘vaccinated patients’ should be quoted. Also, ‘immunocompromised’ should replace ‘vulnerable’ patients.

Response: We thank reviewer 2 for those suggestions to improve the wording. We have replaced in the abstract “patients” by “participants”. Furthermore, we have replaced “immunocompromised patients” by “patients with different levels of immunosuppression” (see revised manuscript, lines 108, 258, and 271). 

Reviewer 3

R3.1.: In the main text it is mentioned that data was collected in 2022-2023, thus it should be clarified that vaccination by 2 doses was not a primary vaccination regimen, as the majority of patients received full doses of vaccine during the pandemic beginning

Response: We agree that this was not entirely clear and have therefore adapted our manuscript accordingly. “In brief, cohort participants from the SHCS and the STCS who have previously already revived the “basic immunization” SARS-CoV-2 vaccination (e.g. two doses of Spikevax from Moderna or two doses of Comirnaty from Pfizer-BioNtech) and who received the bivalent vaccine (mRNA-1273.214 or BA.1–adapted BNT162b2) were recruited from the 27th October 2022 until the 24th January 2023” (see revised manuscript, lines 138-142). Furthermore, the baseline Table contains information about the numbers of previously received SARS-CoV-2 vaccines (Table S1).

R3.2.: In the Methods section it is mentioned, that patients provided samples at several time points, but it seems like not all the patients provided blood samples at all the time points. Could you clarify why? did the timepoint of blood test from vaccination had an impact on the tests result?

Response: The fact that not all patients show up at all study visits is rather common in clinical studies and something which we also observed in all our COVERALL studies. As COVERALL-3 was an observational study we did not systematically assess the reasons for missed study visits. Our analyses assessing the agreement between the SARS-CoV-2 IGRA by Euroimmun and the IGRA SARS‑CoV‑2 by Roche for each time point (Table S4) does not indicate any different results. For more information about the durability of the immune response we have to refer to our main results paper (Amstutz A et al. Antibody and T-cell response to bivalent booster SARS-CoV-2 vaccines in people with compromised immune function (COVERALL-3). J Infect Dis. 2024).

R3.3. Authors should explain how kappa and OPA are calculated, otherwise it is not clear why there is a discrepancy between the kappa and OPA

Response: We clarify now in more detail the different concepts of these statistics (i.e. Kappa takes into account the agreement by chance) and cite relevant references (see revised manuscript, lines 182-187). 

R3.4: the main point that the methodology of the compared tests is different, thus it is hard to compare the results

Response: We agree that the two diagnostic tests use different methodology. Thus, from a researchers’ perspective it might not be surprising that the agreement is not perfect. However, both diagnostic tests are routinely used and are commercially available - to assess if a patient has a T-cell response or not. Hence, we would argue that from a patient and clinician’s perspective it is rather worrisome if two routinely used tests can rather strongly diverge regarding a positive or negative result, a fact clinicians need to take in to account when communicating results to patients 

R3.5.: How do you explain more invalid results with Roche in PWH and more invalid results of Euroimmun in SOT?

Response: To answer this question in more detail, we added two columns to Table S6 and split the invalid results of the Euroimmun and the Roche test into two groups: PWH and SOT recipients. As the reviewer mentions, most invalid results with Euroimmun are invalid positive controls in SOT recipients. This might be a result of the more immunosuppressed state of this patient group compared to PWH, in combination with the antigen used in the Euroimmun test. We added this to the text (see revised manuscript, lines 228-229 and 259-263). In the Roche test, invalid results are mostly invalid negative controls in PWH. We don’t have a good explanation for this phenomenon.

Additional points: 

• With a heavy heart we have to inform you that the first author Annette Audigé has passed away on the 27th September 2024. We have all been working with her very closely and have implemented the requested changes also in her interest, having access to all study related data. We have included the information about the death of Annette Audigé in the authors contribution section.

• As requested by the journal, we provide the data set for the diagnostic comparison (see supplementary excel file).

---

## [Decision Letter · Decision Letter 1]

8 Jan 2025

Low agreement and frequent invalid controls in two SARS-CoV-2 T-cell assays in people with compromised immune function

PONE-D-24-37872R1

Dear Dr. Speich,

We’re pleased to inform you that your manuscript has been judged scientifically suitable for publication and will be formally accepted for publication once it meets all outstanding technical requirements.

Kind regards,

Mao-Shui Wang

Academic Editor

PLOS ONE

Additional Editor Comments (optional):

Reviewers' comments:

Reviewer's Responses to Questions

**Comments to the Author**

1. If the authors have adequately addressed your comments raised in a previous round of review and you feel that this manuscript is now acceptable for publication, you may indicate that here to bypass the “Comments to the Author” section, enter your conflict of interest statement in the “Confidential to Editor” section, and submit your "Accept" recommendation.

Reviewer #1: (No Response)

Reviewer #2: All comments have been addressed

2. Is the manuscript technically sound, and do the data support the conclusions?

Reviewer #1: (No Response)

Reviewer #2: Yes

3. Has the statistical analysis been performed appropriately and rigorously? 

Reviewer #1: (No Response)

Reviewer #2: Yes

4. Have the authors made all data underlying the findings in their manuscript fully available?

Reviewer #1: (No Response)

Reviewer #2: Yes

5. Is the manuscript presented in an intelligible fashion and written in standard English?

Reviewer #1: (No Response)

Reviewer #2: Yes

6. Review Comments to the Author

Reviewer #1: (No Response)

Reviewer #2: Thank you for the point by point response. Positivity thresholds for the tests have been added and the discussion about the antigen has been completed.

7. PLOS authors have the option to publish the peer review history of their article (what does this mean?). If published, this will include your full peer review and any attached files.

Reviewer #1: No

Reviewer #2: **Yes: **Amélie Guihot

---

## [Editor Report · Acceptance letter]

10 Jan 2025

PONE-D-24-37872R1 

PLOS ONE

Dear Dr. Speich, 

I'm pleased to inform you that your manuscript has been deemed suitable for publication in PLOS ONE. Congratulations! Your manuscript is now being handed over to our production team.

Kind regards, 

on behalf of

Dr. Mao-Shui Wang 

Academic Editor

PLOS ONE